# Applying deep learning to a chemistry-climate model for improved ozone prediction

Zhenze Liu<sup>1</sup>, Ke Li<sup>1</sup>, Oliver Wild<sup>2</sup>, Ruth M. Doherty<sup>3</sup>, Fiona M. O'Connor<sup>4,5</sup>, and Steven T. Turnock<sup>4,6</sup>

**Correspondence:** Ke Li (keli@nuist.edu.cn)

Abstract. Chemistry-climate models have developed significantly over the decades, yet they still exhibit substantial systematic biases in simulating atmospheric composition due to gaps in our understanding of underlying processes. Building on deep learning's success in different domains, we explore its application to correct surface ozone biases in the state-of-theart chemistry-climate model UKESM1. Six statistical models have been developed, and the model Transformer outperforms others due to its advanced architecture. A simple weighted ensemble approach is further proved to enhance performance by 14% over the best single model Transformer, reducing RMSE to 0.69 ppb. Applied to future scenarios (SSP3-7.0 and SSP3-7.0-lowNTCF), the UKESM1 shows a larger overestimation of ozone changes by up to 25 ppb compared to present-day conditions. Despite biases, UKESM1 captures the non-linear ozone sensitivity to precursors, with temperature-sensitive processes identified as a dominant contributor to biases. We highlight that simulations of future surface ozone are likely to become less accurate under a warmer climate. Therefore, the bias correction approaches introduced here have substantial potential to improve the accuracy of ozone impact assessments. These methods are also applicable to other chemistry-climate models, which is critical for informing air quality and climate policy decisions.

## 1 Introduction

Global chemistry-climate models are vital for simulating atmospheric composition and its changes by representing the relevant physical and chemical processes in the atmosphere. However, these models face challenges in accurately reproducing observed concentrations of short-lived species, such as ozone (O<sub>3</sub>). Global models typically have coarse spatial resolution, and this limitation hampers the representation of small-scale processes, leading to systematic biases in simulations (Stock et al., 2014; Fenech et al., 2018). There are currently no simple methods to address these issues effectively without increasing model resolution. However, higher resolution significantly increases computational demands. Besides, increasing model resolution does not consistently improve accuracy, sometimes even introducing new biases (Wild and Prather, 2006; Iles et al., 2020).

<sup>&</sup>lt;sup>1</sup>Jiangsu Key Laboratory of Atmospheric Environment Monitoring and Pollution Control, Collaborative Innovation Centre of Atmospheric Environment and Equipment Technology, School of Environmental Science and Engineering, Nanjing University of Information Science and Technology, Nanjing, China

<sup>&</sup>lt;sup>2</sup>Lancaster Environment Centre, Lancaster University, Lancaster, UK

<sup>&</sup>lt;sup>3</sup>School of GeoSciences, The University of Edinburgh, Edinburgh, UK

<sup>&</sup>lt;sup>4</sup>Met Office Hadley Centre, Exeter, UK

<sup>&</sup>lt;sup>5</sup>Department of Mathematics and Statistics, Global Systems Institute, University of Exeter, Exeter, UK

<sup>&</sup>lt;sup>6</sup>University of Leeds Met Office Strategic Research Group, School of Earth and Environment, University of Leeds, Leeds, UK

Moreover, evaluating model performance is challenging due to uncertainties in comparing grid-scale outputs with localized, site-based measurements (Schultz et al., 2017).

Considering these issues, surface ozone simulations in current global chemistry-climate models exhibit notable biases, particularly at regional scales (Turnock et al., 2020). Although large-scale ozone distributions are generally well-captured (Fleming et al., 2018; Griffiths et al., 2021), regional ozone concentrations remain challenging to reproduce, especially at the surface where precursor emissions and surface deposition exert strong influences. The assessment of the Tropospheric Ozone Assessment Report (TOAR) also reported that global models exhibit systematic biases in their surface ozone simulations across all seasons, with a multi-model mean bias of 7.7 ppb (approximately a 20% overestimation) in the Northern Hemisphere (Young et al., 2018). These biases may stem from inadequate representation of dynamics (e.g., meteorology and deposition), and oversimplified ozone chemistry (Archibald et al., 2020a). However, efforts to improve individual modules, such as chemistry schemes, can even result in greater biases in ozone simulation (Archer-Nicholls et al., 2021). Progress in addressing these issues has been limited over recent decades (Revell et al., 2018; Wild et al., 2020).

Deep learning, a transformative approach in fields like computer vision and natural language processing (LeCun et al., 2015), is increasingly applied in physical science (Reichstein et al., 2019). Recent studies have demonstrated its growing use in atmospheric science. It has shown promise in weather modeling and data generation. Specific applications include mimicking atmospheric photochemical processes (Xing et al., 2022), and directly predicting future weather (Bi et al., 2023; Lam et al., 2023), often outperforming traditional numerical methods in speed and accuracy. The uncertain parameterizations e.g., moist physics and radiation processes in climate models can also be replaced by deep learning models (Wang et al., 2022). Another key advantage of deep learning is its ability to fuse multi-source data, enabling the creation of global datasets, such as surface ozone concentrations (Betancourt et al., 2022). However, its application to air pollution modeling, particularly for ozone, is challenging due to the localized nature of pollution and limited observational data for key variables. To address this, we adopt a hybrid approach, integrating process-based chemistry-climate models with deep learning to improve the accuracy of ozone simulations.

Bias correction, as a way to further improve model accuracy, has been developed for different goals. For instance, Vaittinada Ayar et al. (2021) aim to distinguish the impacts of different uncertainties (e.g., emissions, scenario, model designs, etc.) on model biases. Vrac and Friederichs (2015) and Nivron et al., (2025) focus on preserving temporal properties in bias correction, such as the frequency of heatwaves over long periods. Machine learning has also been applied to surface ozone bias correction (e.g., Ivatt and Evans (2020), Miyazaki et al. (2025)), achieving substantial error reduction. However, most studies have not fully explored the performance of different deep learning approaches, and their impacts on prediction remain uncertain. In our work, we therefore explore several deep learning models for ozone bias correction and propose a weighted ensemble to achieve more robust results.

In this study we investigate the potential of deep learning to correct surface ozone biases in a global chemistry-climate model. In Section 2, we describe the chemistry-climate model and introduce six statistical models used for bias correction. Section 3 evaluates their performance and proposes a weighting scheme to optimize results. Section 4 demonstrates the advantages of

this approach for projecting future surface ozone changes. In Section 5, we analyze the sensitivity of ozone in both the original and bias-corrected models. Finally, Section 6 presents our conclusions.

# 2 Approach

## 2.1 Chemistry-climate model and experiments

We use version 1 of the United Kingdom Earth System Model (UKESM1; Sellar et al. (2019)) to simulate present-day (2004–2014) and future (2045–2055) surface O<sub>3</sub> mixing ratios under different emission and climate scenarios. UKESM1 incorporates a physical climate model, the Hadley Centre Global Environment Model version 3 (HadGEM3), configured with the Global Atmosphere 7.1 and Global Land 7.0 (GA7.1/GL7.0; Walters et al. (2019)). Chemistry is simulated using the state-of-the-art United Kingdom Chemistry and Aerosol module (UKCA; O'Connor et al. (2014)), which includes a unified stratosphere–troposphere gas-phase chemistry scheme (StratTrop; Archibald et al. (2020b)). In this study, an extended version of this chemistry scheme incorporating additional reactive volatile organic compounds (VOCs) is employed to improve the representation of O<sub>3</sub> production (Liu et al., 2021). The model resolution is N96L85 in the atmosphere, with 1.875° in longitude by 1.25° in latitude, 85 terrain-following hybrid height layers, and a model top at 85 km. The model is nudged with ERA-Interim reanalyses every 6h for present-day simulations.

In our present-day simulations (2004-2014), we use anthropogenic (Hoesly et al., 2018) and biomass (Van Marle et al., 2017) emissions from the Coupled-Model Intercomparison Project Phase 6 (CMIP6; Eyring et al. (2016)). Biogenic VOC emissions are calculated online in the Joint UK Land Environmental Simulator (JULES) land-surface scheme (Eyring et al., 2016). For future simulations (2045-2055), we use the shared socio-economic pathways (SSP; O'Neill et al. (2014)), which represent various trajectories for emission and climate policies, considering social, economic and environmental development (Rao et al., 2017). We select the SSP3-7.0 and SSP3-7.0-lowNTCF pathways to illustrate the effects of weaker and stronger air pollutant emission controls, respectively. Both pathways anticipate a warmer and more humid climate, although SSP3-7.0-lowNTCF includes significant reductions in anthropogenic emissions of near-term climate forcers (NTCF), such as O<sub>3</sub> precursors and aerosols. Details of the present-day and future emissions under SSP3-7.0 and SSP3-7.0-lowNTCF are provided in Liu et al. (2022b). Other emissions, including sea salt, dust, and lightning NO<sub>x</sub>, are the same as those used in UKESM1 simulations for CMIP6 (Turnock et al., 2020). The atmosphere-only configuration of UKESM1 is applied with prescribed sea surface temperatures and sea ice to examine the transient impacts of emissions under present-day and future climates.

# 2.2 Six approaches for $O_3$ bias correction

Surface ozone concentrations are typically underestimated in winter and overestimated in summer when simulated with UKESM1 (Archibald et al., 2020b). The biases with consistently high values across all seasons, are also observed in other chemistry-climate models used in CMIP6 (Young et al., 2018; Turnock et al., 2020). However, the underlying reasons for these biases in each model remain unclear. Reliable model outputs can still be achieved by bias correction, as long as the systematic

biases are mitigated. Our goal is to correct these biases directly through different statistical and deep learning methods. We assume that these systematics errors are specific to different process-based models but can be learned from historical data using statistical approaches, and further infer how large the biases will be in future scenarios. The model is nudged for historical runs, so the systematic errors would only represent those caused by parameterizations of internal processes, rather than by external data sources such as meteorology.

As a reference dataset for correcting O<sub>3</sub> simulated with UKESM1, we consider surface O<sub>3</sub> reanalysis data from the European Centre for Medium-Range Weather Forecasts (ECMWF) Atmospheric Composition Reanalysis 4 (EAC4) under the Copernicus Atmosphere Monitoring Service (CAMS; Inness et al. (2019)). One advantage of the CAMS reanalysis is its better agreement with TOAR O<sub>3</sub> observations, exhibiting mean seasonal biases of about 3 ppb, notably lower than the biases of up to 16 ppb in UKESM1 at locations where TOAR observations are available (Turnock et al., 2020). A comparison and evaluation of UKESM1, CAMS, and TOAR has been conducted in Liu et al. (2022a). However, we note that CAMS still has biases, especially in regions with sparse observations (e.g., East Asia, Southeast Asia; Huijnen et al., 2020). These limitations may propagate to our corrections; however, CAMS data is still a suitable benchmark for demonstrating our methodology due to its lower biases. In addition, the spatial scale of these data closely aligns with the output of UKESM1, thereby mitigating uncertainties related to the spatial representativeness of sparse observations. We note that the large volume of the dataset, providing global coverage, is crucial for training deep learning models. Future applications of bias correction could be replaced by a measurement-based surface O<sub>3</sub> climatology if this becomes available in future.

100

105

120

Here we apply six approaches to calculate surface O<sub>3</sub> biases. Fig. 1 illustrates the increasing complexity of these methods from left to right, starting with multiple linear regression (MLR), random forest (RF), multilayer perceptron (MLP), convolutional neural network (CNN), residual network (ResNet) and Transformer. MLR is a linear method, while RF transforms linear processes into nonlinear ones through decision tree-based layers. MLP forms the basis of deep learning, incorporating a feed-forward neural network (FFN). CNN uses convolutional operators as encoders, which are particularly effective for processing two-dimensional data, such as images. ResNet is an architecture that enables the training of deep learning models with multiple layers, addressing challenges that were prevalent during the early development of deep learning (He et al., 2016). The Transformer, a more recent architecture, demonstrates strong capabilities in processing long-sequence tasks, such as natural language understanding, with its core functionality driven by the Attention mechanism (Vaswani et al., 2017).

We assume that UKESM1 exhibits systematic biases that are associated with other self-generated variables. The main variables relevant to ozone production and transport are selected as follows (Liu et al., 2022a). We use 20 physical, meteorological, and chemical variables as features, including location, season, temperature, humidity, wind speed, photolysis and deposition rates, and concentrations of key precursors. For MLR, RF, and MLP, the features and O<sub>3</sub> biases corresponding to the same model grid cell are used to train the different approaches. For the methods designed to process 2D data, input pairs consist of a 9x9 grid cell patch centered around the grid cell where O<sub>3</sub> biases are to be calculated. We also calculate the ensemble mean of all models to optimize predictions.

The feature data are obtained from UKESM1 simulations, and surface O<sub>3</sub> biases are derived from the differences between UKESM1 simulations and the CAMS reanalysis. Monthly mean O<sub>3</sub> mixing ratios from the lowest layer in UKESM1 are

Figure 1. The architectures of MLR, RF, MLP, CNN, ResNet and Transformer applied in this study for calculating surface  $O_3$  biases. Each diagram illustrates the workflow, beginning with the input of features to the prediction of  $O_3$  biases. MLR, RF, and MLP receive input features from a single model grid cell (1x1), whereas the remaining models process features from a 9x9 block of grid cells.

used. The dataset is split into 80% for training, 10% for validation, and 10% for testing, with approximately 2.9 million data samples used for model training. We choose mean absolute error as the loss function and AdamW as the optimizer to minimize it. To increase model regularization, a weight decay value of 0.001 is applied to constrain the size of parameter weights. The initial learning rate is set to 0.01, with a cosine annealing schedule for dynamic adjustment of learning rates to improve training (Loshchilov and Hutter, 2016). As the complexity of models increases, so does the number of parameters; however, we limit the number of trainable parameters in our most complex model, the Transformer, to 9 million to manage computational resources. The Transformer requires approximately 8 hours to converge on a single GPU (RTX 3090 Ti).

**Table 1.** Overview of the 20 input features used for model training.

| Category       | Specific Features                                                                                 | Units          |
|----------------|---------------------------------------------------------------------------------------------------|----------------|
| Location       | Latitude, Longitude                                                                               | Degrees        |
| Time           | Month                                                                                             | -              |
| Geography      | Land, Elevation                                                                                   | -, m           |
| Meteorology    | Temperature (at 2m), Pressure, Relative humidity, Wind speed (u/v components)                     | K, hPa, %, m/s |
| Photolysis     | $j(NO_2), j(O^1D)$                                                                                | $s^{-1}$       |
| Deposition     | Dry deposition velocity of O <sub>3</sub>                                                         | m/s            |
| Chemistry      | NOx (NO+NO <sub>2</sub> ), VOCs (sum of non-methane species), Isoprene, OH, PAN, HNO <sub>3</sub> | $\mu$ g/m $^3$ |
| Boundary Layer | Boundary layer height                                                                             | m              |

## 3 Statistical model evaluation and the weighting scheme

The performance of all 6 statistical models is evaluated using testing data to give an independent assessment, see Fig. 2. All models generally simulate the surface ozone biases in UKESM1 effectively, capturing both underestimations and overestimations. However, the deep learning models (Fig. 2c–f) clearly outperform the simpler linear and random forest models (Fig. 2a, b). A primary limitation of the linear and random forest models is their inability to capture extreme bias values, with many predictions clustering around 0 ppb. Overall, the systematic biases are smoothly distributed with a mean near 0, indicating that underestimations and overestimations occur with comparable frequency in UKESM1.

Both the ResNet and Transformer approaches perform best, with their predictions closely aligning with the 1:1 line across the full range of biases. These models yield higher correlation coefficients (up to 0.997) and lower root-mean-square errors (RMSE). From MLP to Transformer, the error is reduced by 64% from 2.25 ppb to 0.8 ppb, highlighting the importance of architecture in this task. However, the improvement from convolution-based models (CNN and ResNet) to the Transformer is marginal. In the deep learning field, the optimal architecture for processing 2D data, whether convolution-based or attention-based, remains a subject of ongoing debate (Smith et al., 2023).

Figure 2. Evaluation of the models' performance in simulating monthly mean surface  $O_3$  biases at each UKESM1 grid point, based on testing data. (a) Surface  $O_3$  biases (UKESM1 minus CAMS) and biases predicted by the models. (b) Probability density function of surface  $O_3$  biases (labelled as "Reference") and the predicted  $O_3$  biases. Statistics are shown in the top-right corner of each panel.

Given that we employ a variety of models, it is logical to consider combining them to reduce the uncertainties inherent in each. Previous studies have demonstrated that integrating multiple models can effectively decrease both uncertainties and prediction errors (Stevenson et al., 2006). However, assigning weights to each model based on their respective performances can produce a more robust outcome compared to simple averaging (Amos et al., 2020). Therefore, we adopt a simple weighted ensemble mean scheme, following the approach outlined by Amos et al. (2020). The calculation of the weights for each model

Figure 3. RMSE of the weighted-mean model in simulating surface  $O_3$  biases as a function of the tuned parameter  $\sigma$ . The best-performing single model, Transformer, is indicated for comparison. The weights assigned to each model corresponding to the optimal value of  $\sigma$  are provided in the text. The sigma values are binned on a linear scale separately into the following ranges: 0.001–0.01, 0.01–0.1, 0.1–1, and 1–10.

i is presented as follows:

$$w_i = \frac{\exp\left(-\frac{D_i^2}{N_i \sigma^2}\right) \times 100}{\sum_i \exp\left(-\frac{D_i^2}{N_i \sigma^2}\right)} \tag{1}$$

Here,  $D_i^2$  represents the squared error between the predictions of an individual model and the reference data, derived from the testing data.  $N_i$  denotes the number of testing data points. The parameter  $\sigma$  is adjustable and can be optimized to determine the most effective weight values. As illustrated in Fig. 3, the error of the weighted-mean model is lower than that of any single model, including the best-performing single model, Transformer, which exhibits an error of 0.80 ppb. The optimal value of  $\sigma = 0.35$  corresponds to the lowest error of the weighted-mean model (0.69 ppb), resulting in a 14% improvement over the Transformer model. We note that the optimal value of  $\sigma$  may differ across various model ensembles. High-performing models, such as ResNet and Transformer, are assigned large weights, approximately 40% each, while the CNN model has a weight of 17%. Models with low performance are excluded due to their limited contribution. This demonstrates that a simple weighting scheme can effectively integrate the outputs of all models, and further improve prediction accuracy. The optimal weighted-mean predictions are used for subsequent analyses.

# 4 Improved assessment of future changes in surface O<sub>3</sub>

Considering the expected biases in future simulations of surface  $O_3$  using UKESM1, we employ deep learning models to predict these biases based on input variables generated from UKESM1 future simulations. Subsequently, a bias-corrected surface  $O_3$  concentration is derived by subtracting the  $O_3$  bias from the simulated  $O_3$  values. Fig. 4 illustrates seasonal variations in weighted-mean surface  $O_3$  concentrations under SSP3-7.0 and SSP3-7.0-lowNTCF scenarios. Compared with bias-corrected results, UKESM1 simulations demonstate much higher global mean  $O_3$  concentrations in summer and similar levels in winter (Fig. 4a, d, g, j). This indicates that the UKESM1 has a greater sensitivity of seasonal  $O_3$  changes, showing a 12 ppb increase compared to the corrected 5 ppb. Higher emissions of  $O_3$  precursors under SSP3-7.0 lead to higher surface  $O_3$  mixing ratios compared to SSP3-7.0-lowNTCF, with differences of 4 ppb in summer and 1.5 ppb in winter (Fig. 4b, e, h, k). In addition, seasonal  $O_3$  variation (winter to summer) becomes more pronounced under SSP3-7.0 (4.4 ppb increase; Fig. 4b, e) than under SSP3-7.0-lowNTCF (2.0 ppb increase; Fig. 4g, h), which is also observed in UKESM1 simulations. Decreased  $O_3$  titration by NO in winter and lower photochemical  $O_3$  production in summer in the lower-emission scenario will both contribute to a reduced seasonal variation.

Fig. 4c, f, i, l shows the changes in surface  $O_3$  from the present day to the future, as simulated by the bias-corrected weighted-mean model. It reveals that distinct emission pathways result in divergent  $O_3$  responses. Under SSP3-7.0, surface  $O_3$  mixing ratios exhibit a consistent increase across both seasons, whereas under SSP3-7.0-lowNTCF, a decrease is simulated. However, the magnitude of  $O_3$  responses is greater under SSP3-7.0-lowNTCF compared to SSP3-7.0. At regional scales, substantial reductions in surface  $O_3$  are shown in North America during summer (Fig. 4c, i), attributable to lower precursor emissions in both scenarios. In contrast, in East Asia, surface  $O_3$  changes vary markedly between scenarios and seasons, driven primarily by differing  $O_3$  chemical environments due to the current high local emissions. These variations pose significant challenges for addressing regional air pollution. Additionally, we compare surface  $O_3$  changes with and without bias correction. While the direction of surface  $O_3$  changes remains generally consistent across most continental regions, opposing signs emerge in certain oceanic areas. This discrepancy may stem from the limited availability of observational constraints in oceanic regions, which hinders both the development of process-based models and the reliable reference data for bias correction. Overall, the influence of different emission pathways on future  $O_3$  concentrations are certain at large scales, particularly over land areas.

In Fig. 5, we further show regional surface  $O_3$  changes from the present day to the future, and compare the predictions of UKESM1 with those derived from the bias-corrected weighted-mean model. Under both future scenarios, surface  $O_3$  changes in most geographical regions fall in quadrants where the signs of the changes are the same, indicating that the effects of emission changes on future  $O_3$  are generally robust. However, in the wintertime, there are differences in sign, especially in high-emission regions such as Asia (Fig. 5a and b) and North America (Fig. 5b). This suggests that the response of  $O_3$  to its precursors, particularly in high- $NO_x$  environments in winter is not well represented in current models. In contrast, there is broad agreement in the sign of  $O_3$  changes in the summertime.

While the sign of O<sub>3</sub> changes is generally consistent between UKESM1 simulations and bias-corrected predictions, the magnitudes of these changes differ substantially. Under the SSP3-7.0 scenario (Fig. 5a), surface O<sub>3</sub> increases in most regions are

**Figure 4.** Comparison of UKESM1 simulated surface O<sub>3</sub> mixing ratios (a, d, g, j) with weighted-mean bias-corrected results (b, e, h, k), and bias-corrected O<sub>3</sub> changes (c, f, i, l) from present day (PD; 2004–2014) to future (2045–2055) under SSP3-7.0 and SSP3-7.0-lowNTCF scenarios. Shown for June-July-August (JJA) and December-January-February (DJF), with hatched regions denoting where the sign of bias-corrected O<sub>3</sub> changes differs from those simulated with UKESM1. Global area-weighted mean mixing ratios are shown in the top-right corner of each panel.

greater in UKESM1 simulations than in bias-corrected estimates, with notably larger overestimations in regions such as North America and Europe during winter, where UKESM1-simulated increases exceed bias-corrected values by more than a factor

**Figure 5.** Seasonal changes in surface O<sub>3</sub> mixing ratios (in ppb) under (a) SSP3-7.0 and (b) SSP3-7.0-lowNTCF scenarios in different global regions, comparing bias-corrected changes with those from UKESM1 simulations. The error bars represent one standard deviation of the surface O<sub>3</sub> changes in the specified region. Markers in light colors denote regions where the magnitudes of biases in UKESM1 present-day simulations rank among the top three for the respective seasons.

of 2. This suggests that UKESM1 may overestimate surface  $O_3$  increases. Similarly, under the SSP3-7.0-lowNTCF scenario (Fig. 5b), surface  $O_3$  decreases in most regions are less pronounced in bias-corrected predictions compared to UKESM1 simulations, indicating an overestimation of  $O_3$  reductions by UKESM1. These findings imply that the impacts of emission and climate policies on surface  $O_3$  concentrations under both scenarios may be smaller than projected by UKESM1 simulations.

195

200

It is acknowledged that large uncertainties remain in these comparisons at regional scales, as the CAMS dataset exhibits substantial biases in certain regions when compared to the TOAR dataset, particularly in East Asia and Southeast Asia (Huijnen et al., 2020). In addition, we also find that there are notable discrepancies between CAMS and UKESM1 especially in regions where observations are unavailable, such as the Middle East (shown as light markers in Fig. 5). Therefore, in these regions exhibiting large biases in UKESM1 simulations, large differences in surface O<sub>3</sub> predictions between UKESM1 and biascorrected UKESM1 also tend to be observed. Bias correction in these regions may lack reliability. Nevertheless, in North

America and Europe, where the CAMS data are more consistent with TOAR observations, with biases of less than 10% (Huijnen et al., 2020), the overestimation of surface O<sub>3</sub> changes by UKESM1 appears more substantiated.

Figure 6. Surface  $O_3$  biases derived from the weighted-mean statistical models for the present day, SSP3-7.0 and SSP3-7.0-lowNTCF scenarios during (a) summer (JJA) and (b) winter (DJF). The biases are presented as a function of the corresponding surface  $NO_x$  mixing ratios (in ppb). The error bars represent one standard deviation of the  $O_3$  biases within each  $NO_x$  bin.

At the global scale, it is evident that UKESM1 simulations consistently overestimate surface  $O_3$  changes during summer (Fig. 6a). In summer, surface  $O_3$  biases peak at approximately 15–30 ppb for  $NO_x$  mixing ratios of 10–15 ppb, typically corresponding to polluted urban areas with large populations (Kephart et al., 2023). The SSP3-7.0 scenario exhibits the largest biases, followed by SSP3-7.0-lowNTCF. Both future scenarios, characterized by high or low emissions, show greater biases (up to 25 ppb) than the present-day scenario, suggesting that emissions are not the primary driver of these larger biases. In contrast, during winter (Fig. 6b),  $O_3$  biases are generally lower. The SSP3-7.0 and SSP3-7.0-lowNTCF biases appear to shift from negative values in the present day to positive or near-zero values. These findings indicate that the underlying biases in surface  $O_3$  simulations are likely to increase under both emission pathways in the future, presenting a challenge to accurately assessing the impacts of future emissions, particularly during summer.

## 215 Sensitivity analysis of surface $O_3$ and $O_3$ biases

Given that the chemical environment affects both the magnitude and sign of surface  $O_3$  changes, it is important for models to accurately represent the non-linear responses of surface  $O_3$  to its precursors. We integrate monthly mean data from all

surface grid cells in both scenarios to derive a relationship between surface  $O_3$  mixing ratios and  $NO_x/VOC$  ratios as simulated by UKESM1, see Fig. 7. Additionally, we show the  $O_3$  sensitivity to the  $NO_x/VOC$  ratio using bias-corrected  $O_3$  data for comparison. The  $NO_x/VOC$  ratio is a simple but effective indicator that distinguishes high- and low- $NO_x$  environments, which reflect different  $O_3$  chemical regimes (Liu et al., 2022b). We calculate  $NO_x$  concentrations by aggregating NO and  $NO_2$  values, and VOC concentrations are calculated by summing the concentrations of all primary emitted non-methane VOC species.

We find that the  $NO_x/VOC$  ratios corresponding to the peaks of surface  $O_3$  concentrations are similar between corrected and uncorrected UKESM1 across different seasons (Fig. 7). The  $NO_x/VOC$  ratio thresholds, which indicate the transition from  $NO_x$ -limited to VOC-limited  $O_3$  production regimes, are higher in summer (1.0-2.0) than in winter (about 0.1). This demonstrates that UKESM1 effectively captures the seasonal variation in critical  $NO_x/VOC$  ratios. The chemical mechanism of UKESM1 accurately represents this transition. In addition, we see that as the  $NO_x/VOC$  ratio increases, the differences between corrected and uncorrected surface  $O_3$  concentrations become more pronounced in summer, but this is less apparent in winter. This suggests that biases in  $O_3$  simulations are amplified under two specific conditions: (1) in regions with high  $NO_x$  levels, such as polluted environments, and (2) in warmer climates, such as during summer. It is noteworthy that  $NO_x/VOC$  thresholds may vary across different chemistry-climate models; however, analyzing  $O_3$  sensitivity to these ratios provides valuable insights into model limitations.

Figure 7. Relationship between surface  $O_3$  mixing ratios (mean per bin) and the  $NO_x/VOC$  ratio (in ppb/ppb) in different seasons, as simulated by UKESM1 and bias-corrected UKESM1. Data are aggregated from monthly means across all global surface grid cells and binned by NOx/VOC ratio. Vertical lines denote the  $NO_x/VOC$  ratios corresponding to the maximum surface  $O_3$  concentrations.

We further investigate the sensitivity of surface  $O_3$  biases to different input variables in the statistical models, usually termed the "feature importance", see Fig. 8. This is calculated as the response of the  $O_3$  bias to a minor perturbation (10%) in each variable, then normalized across all variables and expressed as a percentage. Fig. 8 shows the feature importance of the eight most influential variables. It reveals that temperature is the primary contributor to  $O_3$  biases, associated with the overestimation of  $O_3$  in summer, as demonstrated in Fig. 7. While other variables also play a role, their impacts are substantially less pronounced than that of temperature. This suggests that temperature-sensitive processes are likely the dominant source of  $O_3$  biases in the model.

Other physical variables, including photolysis rates, humidity, boundary layer height and dry deposition, are also associated with surface  $O_3$  biases. Chemical species such as hydroxyl radicals (OH) and peroxyacetyl nitrate (PAN), which are linked to the oxidation of  $O_3$  precursors and regional transport, play a notable role in influencing these biases. While deep learning models highlight the importance of these variables, simpler statistical models, such as MLR and RF, show little sensitivity to them. This suggests that simpler models tend to overemphasize the most dominant variables, whereas complex models may overdistribute feature importance across a broader range of variables. Furthermore, we find that the positive or negative values of feature importance are generally consistent with physical expectations. For example, an increase in the  $NO_2$  photolysis rate,  $j(NO_2)$ , enhances  $O_3$  production and tend to result in higher  $O_3$  biases, which is hence reflected by the positive feature importance of  $j(NO_2)$ . In contrast, an increase in the  $O(^1D)$  photolysis rate,  $j(O_1D)$ , promotes  $O_3$  destruction and leads to lower  $O_3$  biases, which is reflected by its negative feature importance. Although MLR and RF models fail to capture these nuanced relationships, they remain useful for identifying the most influential variables. We highlight that the underlying causes of  $O_3$  biases are complex; however, temperature consistently emerges as the dominant factor, potentially exerting a significant influence on the accuracy of  $O_3$  simulations under future warmer climate conditions.

#### 6 Conclusions

We have successfully applied a range of statistical approaches to correct surface O<sub>3</sub> biases in UKESM1, a state-of-the-art chemistry-climate model. This model typically overestimates surface O<sub>3</sub> concentrations in summer and underestimates them in winter. While these model biases can be corrected using any of the statistical approaches, deep learning models significantly outperform traditional approaches such as multiple linear regression (MLR) and random forest (RF). Among the deep learning architectures, the residual network (ResNet) and Transformer models yield consistent results, with small differences between them. The convolutional neural network (CNN) also produces comparable predictions to ResNet and Transformer. We note that while complex models generally achieve higher prediction accuracy, the full potential of the Transformer architecture may not be fully realized in this task due to the specific nature of the task.

A simple weighted ensemble mean scheme is proposed, demonstrating an additional 14% improvement in performance compared to the best individual approach, the Transformer model. To assess future changes in surface  $O_3$ , we apply bias correction to simulations generated by UKESM1. The signs of surface  $O_3$  changes are generally consistent between corrected and uncorrected UKESM1. However, the magnitudes of these changes differ. Surface  $O_3$  changes simulated by UKESM1 are

Figure 8. The importance of different input features to surface O<sub>3</sub> biases in each statistical model.

typically overestimated in both seasons compared to the bias-corrected changes. Under the SSP3-7.0 scenario, the corrected global summer mean  $O_3$  mixing ratios are projected to increase by 1.2 ppb, whereas under the SSP3-7.0-lowNTCF scenario, they are expected to decrease by 2.8 ppb. In winter, the corrected surface  $O_3$  mixing ratios are projected to increase by 0.5 ppb under SSP3-7.0 and to decrease by 1.1 ppb under SSP3-7.0-lowNTCF.

The sensitivities of surface O<sub>3</sub> to its precursors are also investigated for both UKESM1 and the bias-corrected UKESM1. It reveals that UKESM1 effectively captures the seasonal differences in O<sub>3</sub> sensitivities, as represented by NO<sub>x</sub>/VOC ratios in different seasons. However, under high NO<sub>x</sub>/VOC conditions, UKESM1 notably overestimates O<sub>3</sub> concentrations, particularly during summer. This suggests that under warmer conditions in the future, UKESM1 tends to overestimate O<sub>3</sub> concentrations. This is further confirmed by examining the feature importance for simulated O<sub>3</sub> biases, which identifies temperature as the most important variable influencing these biases. Deep learning models also highlight the importance of other variables; however, their importance is considerably less substantial than that of temperature. This suggests that processes sensitive to temperature variations may have a pronounced influence on O<sub>3</sub> concentrations simulated by UKESM1.

Despite the demonstrated capabilities of deep learning models in capturing surface O<sub>3</sub> biases, we acknowledge that uncertainties remain, particularly regarding the use of CAMS data as a reference for model training. Nevertheless, this exploratory study tests the methodology's feasibility and provides insights into mitigating uncertainties associated with approach selection. It establishes a robust foundation for the broader application of bias correction techniques, particularly through the integration of deep learning with chemistry-climate models. This integration presents a promising pathway for addressing systematic errors in chemistry-climate models, while also facilitating the diagnosis of the underlying causes of model biases. Bias correction techniques stand to gain from the increasing availability of high-quality observational data, with applications extending

beyond  $O_3$  to other atmospheric components. This will strengthen the robustness of assessments in regions where observations are currently lacking, ultimately producing more reliable projections of  $O_3$  changes across different climate scenarios.

Data availability. The data generated in this study are available upon request.

Author contributions. All authors participated in designing the study. ZL conducted the UKESM1 simulations and deep learning analysis with KL. OW, RMD, FMO, and ST provided scientific guidance and interpretation of results. ZL drafted the manuscript, with contributions and revisions from all co-authors.

Competing interests. The contact author has declared that none of the authors has any competing interests.

Acknowledgements. Zhenze Liu thanks the National Natural Science Foundation of China (NSFC), the Natural Science Foundation of Jiangsu Province and the China Postdoctoral Science Foundation for funding under grants 42307140, SBK2023043946 and 2023M731749. Ke Li thanks the National Natural Science Foundation of China for funding under grant 42293323. Oliver Wild and Ruth M. Doherty thank the Natural Environment Research Council (NERC) for funding under grants NE/N006925/1, NE/N006976/1 and NE/N006941/1. Fiona M. O'Connor was supported by the Met Office Hadley Centre Climate Programme funded by BEIS and also acknowledges support from the EU Horizon 2020 Research Programme CRESCENDO (grant agreement number 641816). Steven Turnock would like to acknowledge support from the UK–China Research and Innovation Partnership Fund through the Met Office Climate Science for Service Partnership (CSSP) China as part of the Newton Fund.

- Archer-Nicholls, S., Abraham, N. L., Shin, Y. M., Weber, J., Russo, M. R., Lowe, D., Utembe, S. R., O'Connor, F. M., Kerridge, B., Latter, B., Siddans, R., Jenkin, M., Wild, O., and Archibald, A. T.: The Common Representative Intermediates Mechanism version 2 in the United Kingdom Chemistry and Aerosols Model, Journal of Advances in Modeling Earth Systems, 13, e2020MS002 420, 2021.
  - Archibald, A. T., Neu, J. L., Elshorbany, Y. F., Cooper, O. R., Young, P. J., Wild, O., et al.: Tropospheric Ozone Assessment Report: A critical review of changes in the tropospheric ozone burden and budget from 1850 to 2100, Elementa: Science of the Anthropocene, 8, 34, 2020a.
  - Archibald, A. T., O'Connor, F. M., Abraham, N. L., Archer-Nicholls, S., Chipperfield, M. P., Dalvi, M., Folberth, G. A., Dennison, F., Dhomse, S. S., Griffiths, P. T., et al.: Description and evaluation of the UKCA stratosphere–troposphere chemistry scheme (StratTrop vn 1.0) implemented in UKESM1, Geoscientific Model Development, 13, 1223–1266, 2020b.
  - Betancourt, C., Stomberg, T. T., Edrich, A. K., Schulz, C., Gentner, D. R., and Fritz, S.: Global, high-resolution mapping of tropospheric ozone–explainable machine learning and impact of uncertainties, Geoscientific Model Development, 15, 4331–4354, 2022.
- Bi, K., Xie, L., Zhang, H., Chen, X., Gu, X., and Tian, Q.: Accurate medium-range global weather forecasting with 3D neural networks, Nature, 619, 533–538, 2023.
  - Eyring, V., Bony, S., Meehl, G. A., Senior, C. A., Stevens, B., Stouffer, R. J., and Taylor, K. E.: Overview of the Coupled Model Intercomparison Project Phase 6 (CMIP6) experimental design and organization, Geoscientific Model Development, 9, 1937–1958, 2016.
  - Fenech, S., Doherty, R. M., Heaviside, C., Vardoulakis, S., Macintyre, H., and O'Connor, F. M.: The influence of model spatial resolution on simulated ozone and fine particulate matter for Europe: implications for health impact assessments, Atmospheric Chemistry and Physics, 18, 5765–5784, 2018.
  - Fleming, Z. L., Doherty, R. M., von Schneidemesser, E., Malley, C. S., Cooper, O. R., Pinto, J. P., et al.: Tropospheric Ozone Assessment Report: Present-day ozone distribution and trends relevant to human health, Elementa: Science of the Anthropocene, 6, 12, 2018.
  - Griffiths, P. T., Murray, L. T., Zeng, G., Shin, Y. M., Abraham, N. L., Archibald, A. T., et al.: Tropospheric ozone in CMIP6 simulations, Atmospheric Chemistry and Physics, 21, 4187–4218, 2021.
- He, K., Zhang, X., Ren, S., and Sun, J.: Deep residual learning for image recognition, Proceedings of the IEEE Conference on Computer Vision and Pattern Recognition, pp. 770–778, 2016.
  - Hoesly, R. M., Smith, S. J., Feng, L., Klimont, Z., Janssens-Maenhout, G., Pitkanen, T., et al.: Historical (1750–2014) anthropogenic emissions of reactive gases and aerosols from the Community Emissions Data System (CEDS), Geoscientific Model Development, 11, 369–408, 2018.
- Huijnen, V., Miyazaki, K., Flemming, J., Inness, A., Sekiya, T., and Schultz, M. G.: An intercomparison of tropospheric ozone reanalysis products from CAMS, CAMS interim, TCR-1, and TCR-2, Geoscientific Model Development, 13, 1513–1544, 2020.
  - Iles, C. E., Vautard, R., Strachan, J., Joussaume, S., Eggen, B. R., and Hewitt, C. D.: The benefits of increasing resolution in global and regional climate simulations for European climate extremes, Geoscientific Model Development, 13, 5583–5607, 2020.
- Inness, A., Ades, M., Agustí-Panareda, A., Barré, J., Benedictow, A., Blechschmidt, A.-M., et al.: The CAMS reanalysis of atmospheric composition, Atmospheric Chemistry and Physics, 19, 3515–3556, 2019.

- Kephart, J. L., Gouveia, N., Rodriguez, D. A., Dronova, I., Santos, J. P., et al.: Ambient nitrogen dioxide in 47 187 neighbourhoods across 326 cities in eight Latin American countries: population exposures and associations with urban features, The Lancet Planetary Health, 7, e976–e984, 2023.
- Lam, R., Sanchez-Gonzalez, A., Willson, M., Wirnsberger, P., Fortunato, M., et al.: Learning skillful medium-range global weather forecasting, Science, 382, 1416–1421, 2023.
  - LeCun, Y., Bengio, Y., and Hinton, G.: Deep learning, Nature, 521, 436–444, 2015.

- Liu, Z., Doherty, R. M., Wild, O., O'Connor, F. M., and Turnock, S. T.: Contrasting chemical environments in summertime for atmospheric ozone across major Chinese industrial regions: the effectiveness of emission control strategies, Atmospheric Chemistry and Physics, 21, 10689–10706, 2021.
- Liu, Z., Doherty, R. M., Wild, O., O'Connor, F. M., and Turnock, S. T.: Correcting ozone biases in a global chemistry–climate model: implications for future ozone, Atmospheric Chemistry and Physics, 22, 12543–12557, 2022a.
  - Liu, Z., Doherty, R. M., Wild, O., O'Connor, F. M., and Turnock, S. T.: Tropospheric ozone changes and ozone sensitivity from the present day to the future under shared socio-economic pathways, Atmospheric Chemistry and Physics, 22, 1209–1227, 2022b.
  - Loshchilov, I. and Hutter, F.: SGDR: Stochastic gradient descent with warm restarts, arXiv preprint arXiv:1608.03983, 2016.
- O'Connor, F. M., Johnson, C. E., Morgenstern, O., Abraham, N. L., Braesicke, P., Dalvi, M., et al.: Evaluation of the new UKCA climate-composition model–Part 2: The Troposphere, Geoscientific Model Development, 7, 41–91, 2014.
  - O'Neill, B. C., Kriegler, E., Riahi, K., Ebi, K. L., Hallegatte, S., Carter, T. R., et al.: A new scenario framework for climate change research: the concept of shared socioeconomic pathways, Climatic Change, 122, 387–400, 2014.
  - Rao, S., Klimont, Z., Smith, S. J., Van Dingenen, R., Dentener, F., Bouwman, L., et al.: Future air pollution in the Shared Socio-economic Pathways, Global Environmental Change, 42, 346–358, 2017.
  - Reichstein, M., Camps-Valls, G., Stevens, B., Jung, M., Denzler, J., Carvalhais, N., et al.: Deep learning and process understanding for data-driven Earth system science, Nature, 566, 195–204, 2019.
  - Revell, L. E., Stenke, A., Tummon, F., Peter, T., Chiodo, G., Sukhodolov, T., et al.: Tropospheric ozone in CCMI models and Gaussian process emulation to understand biases in the SOCOLv3 chemistry–climate model, Atmospheric Chemistry and Physics, 18, 16 155–16 172, 2018.
- Schultz, M. G., Schröder, S., Lyapina, O., Cooper, O. R., Galbally, I., Petropavlovskikh, I., et al.: Tropospheric Ozone Assessment Report: Database and metrics data of global surface ozone observations, Elementa: Science of the Anthropocene, 5, 58, 2017.
  - Sellar, A. A., Jones, C. G., Mulcahy, J. P., Tang, Y., Yool, A., Wiltshire, A., et al.: UKESM1: Description and evaluation of the UK Earth System Model, Journal of Advances in Modeling Earth Systems, 11, 4513–4558, 2019.
  - Smith, S. L., Brock, A., Berrada, L., and De, S.: Convnets match vision transformers at scale, arXiv preprint arXiv:2310.16764, 2023.
- Stevenson, D. S., Dentener, F. J., Schultz, M. G., Ellingsen, K., van Noije, T. P. C., Wild, O., et al.: Multimodel ensemble simulations of present-day and near-future tropospheric ozone, Journal of Geophysical Research: Atmospheres, 111, D08 301, 2006.
  - Stock, Z., Russo, M., and Pyle, J. A.: Representing ozone extremes in European megacities: the importance of resolution in a global chemistry climate model, Atmospheric Chemistry and Physics, 14, 3899–3912, 2014.
- Turnock, S. T., Allen, R. J., Andrews, M., Bauer, S. E., Deushi, M., Emmons, L., et al.: Historical and future changes in air pollutants from CMIP6 models, Atmospheric Chemistry and Physics, 20, 14 547–14 579, 2020.
  - Van Marle, M. J., Kloster, S., Magi, B. I., Marlon, J. R., Daniau, A.-L., Field, R. D., et al.: Historic global biomass burning emissions for CMIP6 (BB4CMIP) based on merging satellite observations with proxies and fire models (1750–2015), Geoscientific Model Development, 10, 3329–3357, 2017.

- Vaswani, A., Shazeer, N., Parmar, N., Uszoreit, J., Jones, L., Gomez, A. N., et al.: Attention is all you need, Advances in Neural Information

  Processing Systems, 30, 2017.
  - Walters, D., Baran, A. J., Boutle, I., Brooks, M., Earnshaw, P., Edwards, J., et al.: The Met Office Unified Model global atmosphere 7.0/7.1 and JULES global land 7.0 configurations, Geoscientific Model Development, 12, 1909–1963, 2019.
  - Wang, X., Han, Y., Xue, W., Yang, G., and Zhang, G. J.: Stable climate simulations using a realistic general circulation model with neural network parameterizations for atmospheric moist physics and radiation processes, Geoscientific Model Development, 15, 3923–3940, 2022.

- Wild, O. and Prather, M. J.: Global tropospheric ozone modeling: Quantifying errors due to grid resolution, Journal of Geophysical Research: Atmospheres, 111, D11 305, 2006.
- Wild, O., Voulgarakis, A., O'Connor, F., Lamarque, J.-F., Ryan, E. M., Lee, L., et al.: Global sensitivity analysis of chemistry–climate model budgets of tropospheric ozone and OH: exploring model diversity, Atmospheric Chemistry and Physics, 20, 4047–4058, 2020.
- Xing, J., Zheng, S., Li, S., Ding, A., Qian, Y., Huang, X., et al.: Mimicking atmospheric photochemical modeling with a deep neural network, Atmospheric Research, 265, 105 919, 2022.
  - Young, P. J., Naik, V., Fiore, A. M., Gaudel, A., Guo, J., Lin, M. Y., et al.: Tropospheric Ozone Assessment Report: Assessment of global-scale model performance for global and regional ozone distributions, variability, and trends, Elementa: Science of the Anthropocene, 6, 10, 2018.