# Peer review of "Applying deep learning to a chemistry-climate model for improved ozone prediction"

_EGUsphere, 2025_

## Author Comment (AC1)

Dear Editor and Reviewers:

We thank you for your valuable time and constructive comments on our manuscript. Point-by-point responses are given; the reviewers' comments shown in black bold, our responses in black, and the modified text in blue italic.

Responses to Reviewer 1:

1. **Surface ozone (O3) as simulated in CMIP6 models, like UKCA/UKESM, tend to show large biases against observations (e.g., Turnock et al., 2020). In particular these models tend to simulate too much O3 in the Northern Hemisphere (NH) summer months (JJA) and too little in the NH winter months (DJF). The reasons behind these are not completely understood but likely reflect biases in the models climate, chemistry and emissions (obviously).**

   **Liu et al use a range of techniques -- from very simple statistical techniques to increasingly complex machine learning (ML) ones -- to bias correct UKCA modelled surface O3 during the historic period using "observations" from the Copernicus Atmosphere Monitoring Service (CAMS) reanalysis. CAMS itself is not perfect in this regard as it is model data but does provide globally gridded data at a comparable resolution to the UKCA outputs.**

   **They find that the Transformer (an attention based ML model) performs best and multiple linear regression performs worst. They pool the results of their bias correction models using an ensembling technique and then then use their ensemble bias correction model to bias correct future surface O3 projections. Using this they find that the UKCA global mean surface O3 in JJA under the SSP-370 is overestimated by more than 12 ppb.**

   **I think the results are interesting, but not surprising, the methodology is clear to follow, but not state-of-the-art, and overall this is a well written and thought provoking study. I have my reservations on the reliability of the results but I think this is a publishable study.**

   We appreciate the reviewer's positive comments.

2. **Main concerns:**

   **Nudging -- as I understand it the model simulations are not nudged and as a result one source of the O3 disagreement in UKCA is from the incorrect simulation of climate. Is this true? I think it would be worth making this clearer. Temperature pops out as an important feature and presumably bias correcting for temperature still does nothing for the O3 bias (as shown before**

**in Archibald et al. 2020).**

We agree with the reviewer that the ozone biases might come from an incorrect simulation of the meteorology. However, in our historical simulations, temperature and winds in this model are nudged with ERA-Interim reanalyses every 6 h (Archibald et al. 2020; Liu et al., 2021). In this study, UKESM1 is only free-running in the future scenario simulations. Therefore, meteorological errors are not a major source of the present-day ozone bias. To clarify this, we have now added the following sentence to the model description section.

Line 67:
*" ... 85 terrain-following hybrid height layers, and a model top at 85 km. The model is nudged with ERA-Interim reanalyses every 6 h for present-day simulations."*

Regarding temperature as a key feature in our bias correction, we note that temperature itself is not being directly corrected. Instead, temperature is an important proxy variable for ozone biases. We cannot simply adjust temperature post hoc (because temperature depends on many processes), but by including temperature (and other variables) in our model, we capture how temperature-related processes contribute to ozone bias. We have highlighted this point in the Section 5 'Sensitivity analysis of surface $O_3$ and $O_3$ biases' that temperature-dependent processes should be given more attention.

Line 240:
*"This suggests that temperature-sensitive processes are likely the dominant source of $O_3$ biases in the model."*

Line 252:
*"We highlight that the underlying causes of $O_3$ biases are complex; however, temperature consistently emerges as the dominant factor, potentially exerting a significant influence on the accuracy of $O_3$ simulations under future warmer climate conditions."*

3. **Links to bias correction literature -- on my reading it seems like there is little to no attempt to discuss and place this work in the context of the very broad literature on bias correction, especially for climate models. I would like to know how techniques such as those by Ayar et al. (2021), Vrac and Friederichs (2015) or Nivron et al. (2025) complement, challenge or reinforce the choices of the approach of the task of bias correction taken here.**

Thank you for highlighting this gap. We have now placed our work in the context of the broader bias-correction literature and added a new paragraph in the Introduction.

Line 44:

*"Bias correction, as a way to further improve model accuracy, has been developed for different goals. For instance, Vaittinada Ayar et al. (2021) aim to distinguish the impacts of different uncertainties (e.g., emissions, scenario, model designs, etc.) on model biases. Vrac and Friederichs (2015) and Nivron et al., (2025) focus on preserving temporal properties in bias correction, such as the frequency of heatwaves over long periods. Machine learning has also been applied to surface $O_3$ bias correction (e.g., Ivatt and Evans (2020), Miyazaki et al. (2025)), achieving substantial error reduction. However, most studies have not fully explored the performance of different deep learning approaches, and their impacts on prediction remain uncertain. In our work, we therefore explore several deep learning models for $O_3$ bias correction and propose a weighted ensemble to achieve more robust results."*

**4. CAMS data -- I think the caveat of the use of these data needs to be made more clearly and centrally.**

We agree that the limitations of the CAMS reanalysis data deserve clearer discussion. CAMS is an assimilation product (Inness et al., 2019), not direct surface observations, and it has known regional biases (Huijnen et al., 2020). We have mentioned this caveat in Sections 2.2 and 4. At the same time, we have examined the relative errors: CAMS ozone exhibits much smaller biases than UKESM1 when compared to observational data (TOAR). These comparisons suggest that CAMS provides a reasonable benchmark for bias correction despite its limitations, and avoids the major gaps in spatial coverage and uncertainty in representativeness associated with use of observations directly.

We have updated the text in Section 2.2 to justify the use of CAMS, and we added Figure 1 below to demonstrate the lower errors of CAMS relative to TOAR.

[Figure]

**Figure 1.** Comparison of seasonal mean (December-January-February (DJF) and June-July-August (JJA)) and annual mean surface $O_3$ mixing ratios between (a-c) UKESM1

and TOAR, (d-f) CAMS and TOAR, and (g-i) UKESM1 and TOAR, all averaged over 2004-2014. Global area-weighted average surface mixing ratios (ppb) are shown in the top right of each panel.

Line 91:

*"As a reference dataset for correcting $O_3$ simulated with UKESM1, we consider surface $O_3$ reanalysis from the Copernicus Atmosphere Monitoring Service (CAMS; Inness et al., 2019). One advantage of the CAMS reanalysis is its better agreement with TOAR $O_3$ observations, exhibiting mean seasonal biases of about 3 ppb, notably lower than the biases of up to 16 ppb in UKESM1 at locations where TOAR observations are available (Turnock et al., 2020). A comparison and evaluation of UKESM1, CAMS, and TOAR has been conducted in Liu et al. (2022b). However, we note that CAMS still has biases, especially in regions with sparse observations (e.g., East Asia, Southeast Asia; Huijnen et al., 2020). These limitations may propagate to our corrections; however, CAMS data is still a suitable benchmark for demonstrating our methodology due to its lower biases. In addition, the spatial scale of these data closely aligns with the output of UKESM1, thereby mitigating uncertainties related to the spatial representativeness of sparse observations. We note that the large volume of the dataset, providing global coverage, is crucial for training deep learning models.*  *Future applications of bias correction could be replaced by a measurement-based surface $O_3$ climatology if this becomes available in the future."*

**5. More general:**

**It is stated that you use 20 features in training your models but you don't articulate exactly what these are. Please can you add a table.**

Thank you for this suggestion. The 20 features are listed briefly in Section 2.2, but a detailed table is now provided in Section 2.2, as shown below.

Table 1: Overview of the 20 input features used for model training.

| Category | Specific Features | Units |
|---|---|---|
| Location | Latitude, Longitude | Degrees |
| Time | Month | - |
| Geography | Land, Elevation | -, m |
| Meteorology | Temperature (at 2m), Pressure, Relative humidity, Wind speed (u/v components) | K, hPa, %, m/s |
| Photolysis | $j(NO_2)$, $j(O^1D)$ | $s^{-1}$ |
| Deposition | Dry deposition velocity of $O_3$ | m/s |

| Chemistry | NO$_x$ (NO+NO$_2$), VOCs (sum of non-methane species), Isoprene, OH, PAN, HNO$_3$ | µg/m$^3$ |
|---|---|---|
| Boundary Layer | Boundary layer height | m |

6. **Figure 7 is interesting and the central point about the shift in NOx/VOC emissions regimes changing is clear but could you map out the areas in space where these points are coming from is to provide some spatial context? I am struggling to see how there is only ever one O3 value for a given NOx/VOC ratio (and is this concentrations or emissions? If so be clear on if it's g(N)/g(C) or what the units are).**

Thank you for pointing this out. The points in Fig. 7 represent the aggregated monthly mean data from all global surface grid cells (not single values per ratio), binned by NO$_x$/VOC ratio. The NO$_x$/VOC ratio is defined using concentrations rather than emissions, so that the results are not strongly influenced by transport processes in emission regions. NO$_x$ as NO + NO$_2$ in ppb, VOCs as summed concentrations of primary non-methane VOC species in ppb. We have revised the caption of Figure 7 to clarify this.

Line 251:
*"Figure 7: Relationship between surface O$_3$ mixing ratios (mean per bin) and the NO$_x$/VOC concentration ratio (in ppb/ppb) in different seasons, as simulated by UKESM1 and bias-corrected UKESM1. Data are aggregated from monthly means across all global surface grid cells and binned by NO$_x$/VOC ratio. Vertical lines denote the NO$_x$/VOC ratio values corresponding to the maximum surface O$_3$ concentrations."*

Typically, high NO$_x$/VOC ratios occur in polluted continental regions (North America, Europe, East Asia), as illustrated in Figure 2 below. However, since our focus is on the overall chemical space of O$_3$ sensitivity to NO$_x$/VOC after bias correction, we did not further analyze geographical space for regional details in this study.

[Figure]

**Figure 2:** Spatial distribution of annual mean $NO_x$/VOC ratios in present-day UKESM1 simulations (2004-2014).

**References:**

Archibald, A. T., O'Connor, F. M., Abraham, N. L., Archer-Nicholls, S., Chipperfield, M. P., Dalvi, M., Folberth, G. A., Dennison, F., Dhomse, S. S., Griffiths, P. T., Hardacre, C., Hewitt, A. J., Hill, R. S., Johnson, C. E., Keeble, J., Köhler, M. O., Morgenstern, O., Mulcahy, J. P., Ordóñez, C., Rumbold, S. T., … Zeng, G. (2020). Description and evaluation of the UKCA stratosphere–troposphere chemistry scheme (StratTrop vn 1.0) implemented in UKESM1. Geoscientific Model Development, 13(3), 1223–1266. https://doi.org/10.5194/gmd-13-1223-2020

Liu, Z., Doherty, R. M., Wild, O., Hollaway, M., & O'Connor, F. M. (2021). Contrasting chemical environments in summertime for atmospheric ozone across major Chinese industrial regions: The effectiveness of emission control strategies. Atmospheric Chemistry and Physics, 21(13), 10689–10706. https://doi.org/10.5194/acp-21-10689-2021

Vaittinada Ayar, P., Vrac, M., & Mailhot, A. (2021). Ensemble bias correction of climate simulations: Preserving internal variability. Scientific Reports, 11, 3098. https://doi.org/10.1038/s41598-021-82715-1

Nivron, O., Wischik, D. J., Vrac, M., Shuckburgh, E., & Archibald, A. T. (2025). A temporal stochastic bias correction using a machine learning attention model. Environmental Data Science, 3, e36. https://doi.org/10.1017/eds.2024.42

Vrac, M., & Friederichs, P. (2015). Multivariate—intervariable, spatial, and temporal—bias correction. Journal of Climate, 28(1), 218–237. https://doi.org/10.1175/JCLI-D-14-00059.1

Ivatt, P. D., & Evans, M. J. (2020). Improving the prediction of an atmospheric chemistry transport model using gradient-boosted regression trees. Atmospheric Chemistry and Physics, 20(13), 8063–8082. https://doi.org/10.5194/acp-20-8063-2020

Miyazaki, K., Marchetti, Y., Montgomery, J., Lu, S., & Bowman, K. (2025). Identifying drivers of surface ozone bias in global chemical reanalysis with explainable machine learning. Atmospheric Chemistry and Physics, 25(15), 8507–8532. https://doi.org/10.5194/acp-25-8507-2025

Responses to Reviewer 2:

1. **Liu et al. (2025) uses 6 different statistical models to bias correct surface UKESM1 ozone comparing to CAMS reanalysis. A weighted approach is shown to improve performance over any single model. This bias correction is then applied to future scenarios, specifically SSP3-7.0 and SSP3-7.0-lowNTCF. The manuscript is well written and the figures are broadly of good quality.**

We thank the reviewer for these positive comments.

2. **Major concerns**

   **My major concern with this study is the assumptions behind it and the validity of the bias correction for future scenarios. The authors state "We assume that UKESM1 exhibits systematic biases that are associated with other self-generated variables" (L97), and UKESM biases are then corrected by comparing UKESM1 to CAMS. However, there was only a limited discussion of CAMS, and while CAMS shows reduced biases compared to TOAR observations there were no detailed comments on whether CAMS uses the same emissions as UKESM or how emissions and how these are represented in models could influence biases against observations. How much is CAMS constrained by data assimilation compared to a free-running model such as UKESM, and is correcting ozone for e.g. biases in temperature a valid and fair approach? I feel that more reasoning is required here, along with clear caveats on the approach taken.**

Thank you for raising this important point. We agree that using CAMS as a reference dataset requires careful discussion. CAMS is not raw ground observations but an assimilation product. Nonetheless, CAMS shows much smaller ozone biases than UKESM1 when compared to independent observations from TOAR (see Figure 1 in our response to Reviewer 1). This confirms that it is an appropriate benchmark for demonstrating our approach. We have discussed the performance of CAMS data in Section 2.2 and Section 4. An expansion of the discussion has now been added in Section 2.2 (see the response to point 4 of Reviewer 1).

Another key point is that CAMS serves as a good training reference: the better the reference quality, the more effective the bias correction. If a better dataset became available, it could replace CAMS in this framework. In practice, CAMS provides global coverage at an appropriate spatial scale, unlike sparse station networks, and this is important for training a global model.

Ivatt and Evans (2020) use the XGBoost machine learning method to predict the model biases of GEOS-Chem, and they also found large biases; they used surface observations

(e.g., EPA, EMEP, and GAW) as reference data to train the model. But the coverage of these datasets is limited in certain regions, unlike CAMS that provides global coverage. This would introduce new uncertainties that we have discussed in Section 2.2 - "uncertainties related to the spatial representativeness of sparse observations".

In addition, UKESM1 is nudged under present-day conditions, using the best-known emissions and meteorology, so its biases primarily reflect internal model processes (e.g., chemistry, parameterizations) rather than meteorological conditions.

3. **I am currently also unconvinced that bias-correcting to CAMS for the present day, then applying this bias correction to future scenarios is also valid and fair. How sure are we that both UKESM and CAMS have the correct internal relationships in the present day simulations to be sure that projecting this bias correction into the future, with a different climate state and different emissions, would lead to that bias correction still being valid? Again, I feel that a greater discussion here of the validity of this method and the caveats in doing so should be made, especially as quite strong statements are made assuming that this is an entirely valid approach. The authors state that "This indicates that the UKESM1 has a greater sensitivity of seasonal O3 changes due to unknown reasons" (L149) in the discussion of future surface O3 changes, but these unknown reasons may make this approach less certain to succeed. How sure are the authors that they have the correct assumptions in the calculation of the biases in the present day, and is UKESM the correct model to consider regional air pollution in the context of their wider questions?**

   **Although at the end of the paper, the authors do state that "we acknowledge that uncertainties remain, particularly regarding the use of CAMS data as a reference for model training" (L262-3) I would have preferred a much more detailed discussion of the assumptions and limitations of this study, as from the current manuscript I am not convinced that this is a valid approach. In terms of a technical peice of work it is well formulated and presented, but scientifically I am nervous about the strength of the scientific statements that the authors make, particularly around the size of the biases UKESM may simulate when considering future climates.**

We thank the reviewer for this suggestion. We clarify that we do not directly compare UKESM1 and CAMS as models. Instead, we use CAMS purely as a reference to estimate UKESM1's systematic biases under present-day conditions. Our machine learning model learns the systematic biases from the CAMS comparison under current emission and climate conditions, and then infers the future correction to future UKESM1 simulations. In other words, we aim to correct the model's systematic errors under future scenarios, rather than correcting future emission or climate uncertainties.

A recent work (Miyazaki et al. 2025) also uses a chemical reanalysis dataset from a data assimilation system to train the machine learning model to correct biases in surface $O_3$ simulations. They also succeeded in reproducing the spatiotemporal patterns of $O_3$ concentrations compared with TOAR observations. We have emphasized our purpose in Section 2.2 and also added a new paragraph to discuss the technique of bias correction in the Introduction section (see the response to point 3 of Reviewer 1), and also noted "uncertainties associated with approach selection" in Conclusion.

We have now justified the approach, and further discussed the assumptions of this approach in Section 2.2.

Section 2.2, Line 85:
*"However, the underlying reasons for these biases in each model remain unclear. Reliable model outputs can still be achieved by bias correction, as long as the systematic biases are mitigated (Ivatt and Evans (2020), Miyazaki et al. (2025)). Our goal is to correct these biases directly through different statistical and deep learning methods. We assume that these systematics errors are specific to different process-based models but can be learned from historical data using statistical approaches, and further infer how large the biases will be in future scenarios. The model is nudged for historical runs, so the systematic errors would only represent those caused by parameterizations of internal processes, rather than by external data sources such as meteorology.*

Section 4, Line 165:
*"This indicates that the UKESM1 has a greater sensitivity of seasonal $O_3$ changes,  showing a 12 ppb increase compared to the corrected 5 ppb."*

4. **Specific issues**
   **Figure 3 - why do there seem to be discontinuities at 0.01, 0.1, and 1? The behaviour of the curve seems to jump at each of these values of sigma.**

We thank the reviewer for noting this. This is because we bin the x space into ranges of 0.001-0.01, 0.01-0.1, 0.1-1, and 1-10 separately on a linear scale. This means the bin in each x range are different and not continuous, and hence the y values at the boundaries appear to jump. This does not affect the results. We have now clarified this in the figure caption.

Line 148:
*"The weights assigned to each model corresponding to the optimal value of σ are provided in the text. The sigma values are binned on a linear scale separately into the following ranges: 0.001–0.01, 0.01–0.1, 0.1–1, and 1–10."*

5. **Figure 4 - I needed to zoom in quite a lot to be able to see the detail of the**

**hatching. I would recommend making this plot bigger, perhaps a full-page 6x2 rather than the 3x4 currently presented.**

We appreciate the suggestion. However, considering there are several aspects to compare in Figure 4 (raw simulations, corrected simulations, their differences, scenarios, and seasons), a 6 x 2 panel layout might not be appropriate. Also, the hatching is only plotted in the third row and mainly occurs over oceanic areas, as discussed in Section 4. A 4 x 3 layout may solve this problem. We have now chosen to change from 3 x 4 to 4 x 3, and modified the corresponding panel labels in text.

[Figure]

**Figure 4.** Comparison of UKESM1 simulated surface O$_3$ mixing ratios (a, d, g, j) with weighted-mean bias-corrected results (b, e, h, k), and bias-corrected O$_3$ changes (c, f, i, l) from present day (PD; 2004–2014) to future (2045–2055) under SSP3-7.0 and SSP3-7.0-lowNTCF scenarios. Shown for June-July-August (JJA) and December-January-February (DJF), with hatched regions denoting where the sign of bias-corrected O$_3$ changes differs from those simulated with UKESM1. Global area-weighted mean mixing ratios are shown in the top-right corner of each panel.

**6. Figure 6 - given that the error bars (only 1 standard deviation) in winter mostly all straddle 0, can it be said that there are any biases in that season at all from this method?**

The error bars represent the spatial variability of biases, and biases tend to be smaller biases in winter than in summer (most are positive values in Fig. 6a). This is also confirmed by comparison with the TOAR dataset (Figure 1 in point 4 of our response to Reviewer 1). However, there will still be large biases in polluted regions even in winter, though the NO titration effect can shift positive biases to negative biases.